**Comment on "The origin of methane in the East Siberian Arctic Shelf unraveled with triple**
**isotope analysis," by Sapart et al. (2017)**
Katy J. Sparrow and John D. Kessler
Department of Earth and Environmental Sciences, University of Rochester, Rochester, New York
Correspondence email: katysparrow@gmail.com (K.J.S.); john.kessler@rochester.edu (J.D.K.)


**Abstract**
In this comment, we outline two major concerns regarding some of the key data presented in this
paper. Both of these concerns are associated with the natural abundance radiocarbon-methane
($^{14}$C-CH$_4$) data. First, no systematic methodology is presented, nor previous peer-reviewed
publication referenced, for how these samples were collected, prepared, and ultimately analyzed
for $^{14}$C-CH$_4$. Not only are these procedural details missing, but the critical evaluation of them
using gaseous and aqueous blanks and standards was omitted although these details are essential
for any reader to evaluate the quality of data and subsequent interpretations. Second, due to the
lack of methodological details, the source of the sporadic anthropogenic contamination cannot be
determined and thus it is premature for the authors to suggest it was in the natural environment
prior to sample collection. As the natural $^{14}$C-CH$_4$ data are necessary for the authors' stated
scientific objectives of understanding the origin of methane in the East Siberian Arctic Shelf, our
comment serves to highlight that the study's objectives have not been met.



In the article titled, "The origin of methane in the East Siberian Arctic Shelf unraveled with triple isotope analysis," (5 May, p. 2283, doi:10.5194/bg-14-2283-2017), Célia Sapart and coauthors present natural abundance radiocarbon-methane ($^{14}$C-CH$_4$) measurements from Laptev Sea sediments and waters alongside methane concentration and methane stable isotope measurements. The authors then draw conclusions about methane source-sink dynamics operating in this arctic shelf sea based upon these methane data. Two concerns with the $^{14}$C-CH$_4$ data lead us to question whether these data should be used to interpret this natural system.

The first issue is that the method used to collect and prepare the $^{14}$C-CH$_4$ samples is inadequately described by Sapart et al. and there is no quality control data presented. Radiocarbon-methane is not a routine measurement in natural waters because of the challenges associated with sampling and preparing a trace isotope of a trace gas. In the methods section of the article, the authors cite two techniques that relate only to the $^{14}$C-accelerator mass spectrometry (AMS) analysis, while the methodologies used for the sample collection and preparation steps leading up to the $^{14}$C analyses of sediment and seawater samples are absent. The natural $^{14}$C-CH$_4$ content of a sample can be affected by carbon and CH$_4$ added from the materials it encounters and by any contact with the atmosphere, so quality control measures are necessary to ensure that a sample is not significantly contaminated prior to analysis and that any minor contamination (i.e. blank addition) is accounted for in the final results. In the supplement, the authors write that, "None of the reference and blank measurements were abnormal," without presenting any descriptions of or data stemming from these tests. Refereed techniques for collecting and preparing $^{14}$C-CH$_4$ samples from natural waters (Dean et al., 2017; Elder et al., 2018; Kessler and Reeburgh, 2005; Pack et al., 2015; Pohlman et al., 2000; Sparrow and Kessler, 2017) include detailed qualitative and quantitative descriptions of the measures taken to validate their methodologies. These measures include processing blank (methane-free) waters and treating methane-free gas and methane of known $^{14}$C-CH$_4$ content in the same way as samples. As the $^{14}$C-AMS measurement error is typically very low relative to $^{14}$C-CH$_4$ collection and preparation procedures, we can only assume that the error associated with the processes that most greatly affects the precision, sensitivity, and accuracy of the reported $^{14}$C-CH$_4$ signature is unaccounted for by the authors.

The second issue that calls the integrity of this study's $^{14}$C-CH$_4$ data into question is the existence of super-modern sediment and water column samples (approaching 100 times above modern) in the dataset. As the authors correctly reference, elevated $^{14}$C-CH$_4$ has previously been documented in other ocean waters (Kessler et al., 2008), however, the values presented here are up to 27 times higher than any previously reported elevated value. It is suggested in the main text and in the supplement that the source of the "highly enriched $^{14}$C" is anthropogenic and that it existed in the natural environment prior to sampling. We argue that it is premature to suggest an origin of this enriched $^{14}$C, either environmental release or contamination (incurred during sample collection, processing, and/or analysis) when the $^{14}$C-CH$_4$ methodological details, with appropriate standards and blanks, are absent from the article. The possibility that the enriched $^{14}$C was derived from the sampling equipment, vessel, and/or land-based laboratories was largely dismissed by the authors, while we attest that it is a valid option. The authors discount the possibility that their samples were contaminated during the sampling process, "because no radioactive tracers were used during the sampling expeditions." This argument is untenable because the half-life of $^{14}$C is 5730 years, meaning any surface contamination will persist for tens of thousands of years—well beyond the

specific project where it was used. In addition, the authors highlight that, for sediment samples,
"the higher $^{14}$C values correspond to the lower $CH_4$ concentrations," to suggest that a small
amount of radioactive contamination in the environment was added to a variable background of
naturally occurring $CH_4$, which would most greatly affect the $^{14}$C signature of the smallest sized
(lowest $CH_4$ concentration) samples. This may be true, but another scenario that is also valid
using the same logic is that the contamination was added during the $^{14}$C-$CH_4$ sample collection
and/or preparation processes. This relationship was noted for sediment samples, but we are not
informed in the article or supplement on the relationship between $CH_4$ concentration and $^{14}$C-$CH_4$
content for the seawater samples. The lack of a data table containing the specific triple-isotope
information for each $CH_4$ sample, in the article or in a data repository, has the effect of making
this study unnecessarily opaque for a reader attempting to draw conclusions for themselves. The
authors clearly state that additional experiments are necessary to determine the unknown origin of
this isotopic enrichment, however, without that complimentary data, or at least data that proves it
was in the sediments and waters prior to sample collection, its presence invalidates all $^{14}$C-$CH_4$
data presented in this study from contributing to our understanding of methane dynamics in the
Arctic Ocean.
In a recently published study, we demonstrate how useful natural abundance $^{14}$C-$CH_4$
measurements can be towards understanding the role of ancient sources of methane in arctic shelf
seas (Sparrow et al., 2018). Importantly, in this study, we find that the stable isotope ($\delta^{13}$C-$CH_4$)
and dissolved $CH_4$ concentration data, together, would suggest an entirely different (and, we
argue, incorrect) interpretation of this system, which attests to the importance of $^{14}$C-$CH_4$
measurements for investigations into the origins of methane. When conducting natural abundance
$^{14}$C-$CH_4$ studies, it is imperative that we do so using peer review published methods with
appropriate radiocarbon blanks and standards; otherwise, interpretations made from $^{14}$C-$CH_4$ data
are unverifiable and inconclusive.

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
