# Peer review of "Comment on “The origin of methane in the East Siberian Arctic Shelf unraveled with triple"

_Biogeosciences, 2018_

## Short Comment (SC1) · 12 Apr 2018

Sparrow and Kessler raise some valid points regarding the potential for contamination in the Sapart et al. study of marine 14CH4 in the ESAS.

To further demonstrate the vulnerability for 14CH4 contamination (which I agree is not well addressed by Sapart et al.), I would refer the readers to Figure 3 in Dean et al. 2017 (doi:10.1016/j.watres.2017.03.009) - Figure 1 here - where we showed how important the correction for atmospheric 14CH4 contamination is in natural abundance

14CH4 samples. This demonstrates that very small inputs of extreme 14C outliers can have a significant effect on bulk natural abundance 14C samples, and therefore assessing blanks etc are crucial to a study of this nature (although having worked with this lab group myself I feel confident they can comfortably address these questions).

I look forward to the response from Sapart et al., as their study is an important contribution on the subject of methane emissions from the ESAS.

[Figure]

Fig. 1.

---

## Referee Comment (RC1) · J. Pohlman (Referee) · 7 May 2018

The comment by Sparrow and Kessler raises important concerns about the reporting and authenticity of methane radiocarbon (14C) data presented by Sapart et al. In the study, Sapart et al use 14C data from sediment and water column methane collected in the East Siberian Arctic Shelf to constrain the age of the organic matter from which the methane originated. Additional source information is provided with 13C and deuterium analysis of the methane that is interesting and informative without the 14C data, but as Sparrow and Kessler remark, the inclusion of 14C data has the potential to provide

additional insight into methane dynamics. Sapart et al. report exceptionally positive (14C-enriched) radiocarbon values from methane samples collected from sediments and the water column at one of the sites they investigated. They suggest the values represent an anthropogenic source present in the environment. Sparrow and Kessler contest it is a sampling artifact. Resolving this dispute is critical for how the community will utilize this study to interpret methane dynamics in the ESAS, and elsewhere.

The issues of data reporting raised by Sparrow and Kessler are that the procedures for collecting the samples are not sufficiently described, and that quality control data are absent. Their criticisms are meaningful. Given the challenges associated with collecting and measuring samples collected for natural abundance radiocarbon measurements, the authors of the study should be encouraged to submit a response in this regard. The expectations for proper documentation by Sparrow and Kessler are explicit and should receive careful consideration. Additionally, details about the application of the gas source analyzer should be included as this 14C analytical method is less common. What are the limits of detection and to what extent do the data reported in this study approach that? These details were not included in the manuscript.

Addressing whether or not the exceptionally (and unprecedented) 14C-enriched methane values from the non-ebullition site in Buor-Khaya Bay (ID-11) are an environmental contaminant or a sampling artifact is a more challenging task to manage. The authors of the study suggest it is a "local anthropogenic nuclear contribution..." that was "laterally transported...from the coastal terrestrial permafrost," while Sparrow and Kessler argue it is equally (if not more) likely to be a product of the "sampling equipment, vessel, and/or laboratory." Sparrow and Kessler raise excellent points in support of their argument (please refer to the comment) for which the authors should respond. They also ask why the relationship between lower concentration sediment samples showing a greater degree of 14C-enrichment is not also reported for the water column samples.

This reviewer would additionally like to know what mechanism would allow for radioactive contaminant migration from the terrestrial to the permafrost sphere through (impermeable?) permafrost since the dawn of the nuclear age. That is (seemingly) a long way to travel through an uncertain conduit within a relatively short amount of time. Is this hydrological feasible?

An argument provided by the authors but not recognized by Sparrow and Kessler is that the 14C-elevated samples from the sediment and water column samples were not sampled in a similar manner. The sediment samples were drilled from ice in 2011 and the water samples were collected on a ship in 2012. What is the source of contamination affecting both expeditions? Also, samples from other locations sampled with the same equipment during the ship-board expedition and stored in the same manner during were not contaminated. How could contaminants have randomly only affected the samples from one site, especially given the extent of 14C-enrichment reported. A response from Sparrow and Kessler on this issue would be worthwhile and valuable to an objective evaluation of the age-based conclusions drawn from this study.

There is certain value in the 13C and deuterium data; therefore, knowing if the corresponding 14C data are acceptable for scientific interpretation merits debate.

---

## Author Comment (AC1) · 17 May 2018

We thank Dr. John Pohlman (RC1) for his thorough review of our comment on the article by Sapart et al. (2017). Here we attempt to address his question that asks about what explanations there could be for the "14C-enriched" samples being contaminated by collection/preparation procedures in both sediment and seawater samples (but not all samples) from separate land and sea expeditions.

To be clear, we are not arguing one way or the other as far as where the radiocarbon

(14C) contamination came from, whether it was present in the environment or whether it was derived from sample collection and preparation equipment; instead, we state that we do not have enough information to rule out either scenario.

We re-emphasize here that the lack of a data table containing the specific triple-isotope information for each methane (CH4) sample, in the article or in a data repository, has the effect of making this study unnecessarily opaque for a reader attempting to draw conclusions for themselves. Additionally, the scarcity of methods and absence of quality control data for the collection and preparation of the sediment and seawater 14C-CH4 samples prevents readers from having knowledge of what specific commonalities and differences occur in those procedures.

To list some given information from the article that pertains to this topic: Sediment from directly offshore of Tiksi and seawater samples from the shelf edge due north of the Lena River delta were measured to be "14C-enriched"; there is no seawater data from directly offshore Tiksi and no sediment data from the shelf edge. The 14C-enriched sediment samples were collected from an equipment caravan driving from land onto the sea ice during a winter month of 2011 while the 14C-enriched seawater samples were collected from an unnamed ship during summer 2012. Data is presented for four sediment cores offshore of Tiksi, but only the ID-11 core shows the supermodern 14C-CH4 and it appears from Figure 2 that the enrichment is found in all but two of the ca. 20 samples from that core. The 14C-CH4 data for the other three sediment cores from that area (IID-13, IIID-13, and VD-13; collected later, in 2013 (?)), are not 14C-enriched above modern. However, for reasons unknown, unlike the ID-11 core, there is minimal (only two (?)) 14C-CH4 data above ca. 20 m for those other three cores, while there is data presented above 20 m in those cores for the other analytes (CH4 concentration and CH4 stable isotopes). All of the presented 14C-CH4 seawater samples that were collected from stations at the shelf edge are 14C-CH4 enriched. We will end by noting that the range of 14C-CH4 values for the 14C-enriched sediment and seawater is similar (100 to approaching 10,000 percent Modern Carbon).

Extraneous 14C-CH4 contamination added by collection/preparation procedures can come from a host of vectors in the field and the laboratory and we will not attempt to make up scenarios for how the Sapart et al. samples could have been affected. However, an obvious vector is contact with materials and/or work space that have a (known or unknown) history of tracer work, a possibility that the procedural blanks we describe in our original comment help to exclude or identify.
* * *

---

## Referee Comment (RC2) · Anonymous Referee #2 · 1 Jul 2018

The comment by Sparrow and Kessler presents two main criticisms of the Sapart et al (2017) article: 1) There is a lack of detailed descriptions of 14C-related methodology and a possible lack of control and blank tests for the 14CH4 analyses that would be fully representative of all sample handling. 2) The extremely high 14C values observed in some of the samples (up to 9560 pMC) are likely indicative of a 14C-enriched contaminant and associated methodological problems.

After carefully reading the Sapart et al paper (including the supplement), I find both of these criticisms completely valid. The 14CH4 measurement methodology should

indeed be described in detail, but it is not present in either the main article or the supplement. I also agree with Sparrow and Kessler on the point that fully-representative control and blank tests for the 14CH4 measurements are absolutely imperative for this kind of work, and there is no evidence in the article that such tests were done.

In agreement with Sparrow and Kessler, I also find the Sapart et al hypothesis regarding extremely high 14CH4 values (that this is due to environmental releases of nuclear waste in the region) unconvincing. For the affected sediment core (ID-11), it seems unlikely that pore water movement in the sediments is fast enough to cause such a large 14CH4 enrichment from possible nuclear waste releases that happened only in the last few decades. Further, the largest 14C enrichments are observed for the deepest samples, again inconsistent with a surface nuclear waste source. For the water samples ("shelf edge"), again I find it implausible that nuclear waste releases in this region have affected the more remote (and deeper) waters while not affecting near-coastal waters. Sparrow and Kessler's hypothesis to explain the extreme 14C enrichments (contamination arising from sample processing) seems much more likely.

---

## Author Comment (AC2) · 3 Jul 2018

Katy J. Sparrow and John D. Kessler

katysparrow@gmail.com

We would like to thank Anonymous Referee #2 (RC2) for their thorough review of our comment on the article by Sapart et al. (2017). To clarify one subtle point, in our comment we do not hypothesize that the elevated $^{14}C$-$CH_4$ signatures are due to contamination (incurred during sample collection, processing, and/or analysis) rather than environmental release; instead, we argue that it is premature to suggest, as the authors do, either of these origins when the methodological details, with appropriate standards and blanks, are not presented. As any such details and data remain absent, any conclusions derived from these analyses should be, at best, considered preliminary.

---

## Author Response (AR1)

Author response:

We thank the Associate Editor for her comments and have edited the manuscript (yellow highlight) in accordance with the her suggestion. We clarify that we are not hypothesizing that contamination during or after sample collection is what occurred, but rather we are arguing that it is a valid option (along with environmental release), which procedural quality control measures would exclude or identify. We have also added an Abstract, cut from our original cover letter, which should also help to make this position clear while also summarizing the comment.

We have also added reference to the $^{14}$C-CH$_4$ methodology for natural waters developed by Dean et al. (2017), which we neglected to include in the initial submission.

Lastly, in the third paragraph, we include a sentence cut from our response to RC1, which clearly points out that because a data table containing methane concentration, stable isotope, and radiocarbon information for each sample is not in the article, readers are again blocked from drawing independent conclusions related to this data.

Thank you,

Katy Sparrow and John Kessler

[revised manuscript text omitted]